# Biomimetic light-harvesting funnels for re-directioning of diffuse light

Alexander Pieper[1], Manuel Hohgardt[1], Maximilian Willich[1], Daniel Alexander Gacek [1], Nour Hafi[1], Dominik Pfennig[1], Andreas Albrecht[1] & Peter Jomo Walla[1]

Efficient sunlight harvesting and re-directioning onto small areas has great potential for more widespread use of precious high-performance photovoltaics but so far intrinsic solar concentrator loss mechanisms outweighed the benefits. Here we present an antenna concept allowing high light absorption without high reabsorption or escape-cone losses. An excess of randomly oriented pigments collects light from any direction and funnels the energy to individual acceptors all having identical orientations and emitting ~90% of photons into angles suitable for total internal reflection waveguiding to desired energy converters (funneling diffuse-light re-directioning, FunDiLight). This is achieved using distinct molecules that align efficiently within stretched polymers together with others staying randomly orientated. Emission quantum efficiencies can be >80% and single-foil reabsorption <0.5%. Efficient donor-pool energy funneling, dipole re-orientation, and ~1.5–2 nm nearest donor–acceptor transfer occurs within hundreds to ~20 ps. Single-molecule 3D-polarization experiments confirm nearly parallel emitters. Stacked pigment selection may allow coverage of the entire solar spectrum.

[1] Department for Biophysical Chemistry, Institute for Physical and Theoretical Chemistry, University of Braunschweig, Gaussstrasse 17, 38106 Braunschweig, Germany. Correspondence and requests for materials should be addressed to P.J.W. (email: p.walla@tu-braunschweig.de)

Over millions of years, nature has achieved a remarkable efficiency in harvesting diffuse light photons and directing them onto an energy-converting device, the photosynthetic reaction center[1, 2]. These processes occur in light-harvesting pigment protein complexes that consist of about 300 randomly oriented pigments funneling the energy of absorbed photons toward the reaction center via several ultrafast, very efficient energy transfer steps (Fig. 1a). The concept nature teaches us is based on efficient absorption of diffuse light, funneling excitation energy to special pigments, and directing them on very efficient charge separating units. Depending on the actual supply of solar photons, nature achieves close to unity efficiencies in converting photons into a primary charge transfer. In addition, nature developed mechanisms regulating the flow of excitation energy for effective photoprotection of the pigments in varying light conditions.

It is known that the sun supplies the earth with more energy than the yearly human consumption within less than half a day[3]. Thus techniques that enable efficient collection of a portion of this gigantic flow of energy can potentially solve the demand for sustainable energy supply. However, silicon-based solar cells, for example, can never exceed energy conversion efficiencies higher than ~30% (Shockley–Queisser Limit[4, 5]), since their energy band gap corresponds to photon energies much lower (~800–1200 nm[6]) than photon energies in the maximum of the sunlight spectrum (~500–600 nm). While other photovoltaic materials with much higher energy conversion efficiencies such as InGaP exist for this spectral range[7], they are, unfortunately, extremely expensive[8]. Therefore, they are often used in combination with solar light concentrators that allow collecting light from larger areas and directing it onto much smaller areas of the costly material. However, these solar light concentrators are in most cases based on conventional lens optics that require direct solar irradiation and often active tracking systems for optimal incident irradiation angles[9–11]. They cannot collect diffuse solar light irradiation occurring in cloudy weather conditions or in shady parts of buildings, i.e., the minimum power supply that can be

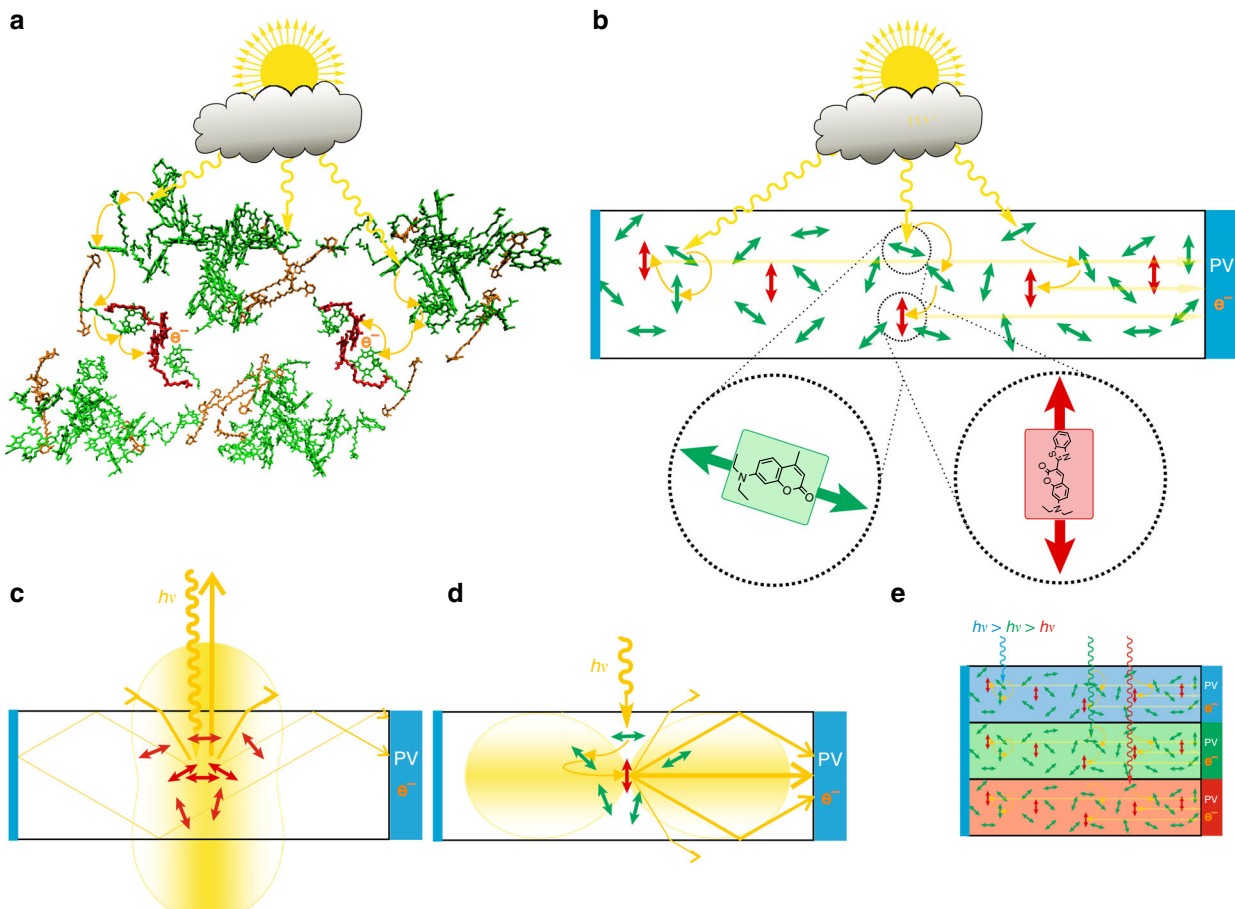

**Fig. 1** Natural and artificial light harvesting. **a** Natural light-harvesting pigment protein complexes absorb diffuse light incident from any direction and efficiently funnel the energy via several ultrafast steps to special pigments converting the energy into a charge separation. Structural data taken from Tanaka et al.[60] and visualized with VMD (Visual Molecular Dynamics)[61]. **b** Artificial light harvesting by several randomly oriented, light-absorbing donor pigments (green) funneling the energy to individual acceptor molecules (red) that all have the same orientation with respect to the laboratory frame. As light is preferentially re-emitted perpendicular to the acceptor dipole moments (red), this allows efficient re-direction of the photons into angle ranges favorable for high efficient photovoltaics (PV) and cost-effective total internal reflection waveguiding (funneling diffuse light redirection, FunDiLight). **c** In classical solar concentrator architectures, molecules are excited that emit preferentially back in directions parallel to the excitation (Photoselection). In addition, full light absorption needs high pigment concentrations inevitably causing reabsorption losses by the same pigments. **d** In contrast, FunDiLight still allows absorbing >99% of donor wavelength light but with much lower acceptor emission re-absorption and ~90% photon re-direction into angles suitable for total internal reflection waveguiding. **e** The concept also allows for stacked structures redirecting light to high efficient photovoltaics for each spectral range. Blue bars on the left in **b**–**e** indicate mirrors

guaranteed under low light conditions is small[10, 11]. For the collection of diffuse light, solar concentrators based on fluorescing pigments have been proposed that re-direct absorbed photons toward photovoltaic devices using total internal reflection[12, 13]. However, full harvesting of solar light would require high concentrations of the pigments that intrinsically lead to multiple re-absorption losses[14–17] as well as aggregation quenching[18–20]. In addition, photoselection of the absorbing pigments and air/concentrator light refraction leads to preferential excitation of horizontal molecules that emit much of the light in unfavorable directions parallel to the incoming light (Fig. 1c) instead of angles suitable for total internal reflection waveguiding. Besides reabsorption, these photon escape-cone losses often dominate all other loss mechanisms in conventional luminescent solar concentrators (LSCs). For randomly oriented molecules and considering photoselection, 30% of the light is lost due to escape-cone losses and the percentage for perfect isotropic emission is only a few percent smaller[21].

Capturing light re-emitted under these unfavorable angles requires expensive dichroic coatings[22–29] and causes multiple reflections and reabsorption losses as well as inefficient incident angles at the photovoltaic devices[30, 31]. To prevent inefficient angles, LSCs have also been proposed that are based on pigments with purposely oriented transition dipole moments enabling direction of the light into much more favorable angle ranges. However, these were based on complex orientation mechanisms—such as electric field alignment with liquid crystals[32–37]—for which it is difficult to achieve sufficient absorbing optical path lengths and which are also not very cost-effective. In addition, when all absorbing dipole moments are oriented nearly perpendicular to the incident irradiation absorption requires even higher concentrations and the problem of reabsorption in the desired emission angles becomes even more severe. Re-absorption itself can be greatly reduced by using dendritic donor–acceptor systems but as they emit the harvested light in all directions still unfavorable emission angles occur[38–52]. It is obvious that a concept that aims at achieving overall absorption and redirection efficiencies close to unity needs to be at least theoretically close to 100% efficient in each individual step, including absorption, energy funneling, and directing the photons toward a photovoltaic device.

Here we present a biomimetic LSC concept and report on findings on how to construct such a concentrator that addresses all these limitations in a single and simple device. The LSC is based on a similar donor–acceptor dye funneling system as natural light-harvesting pigment protein complexes (Fig. 1a, b). It consists of a larger amount of donor pigments that absorb light and funnel it onto few acceptor molecules. However, while the donors are randomly oriented—allowing absorbing nearly the entire light from any incident angles—the acceptor molecules are of well-defined orientation with respect to the laboratory frame. Because of the common orientation, all acceptors emit light in certain predefined directions perpendicular to their transition dipole moment and can therefore direct the light onto an efficient photoconversion device using, for example, very efficient total reflection waveguiding (Fig. 1d). Since only few acceptor molecules direct the light onto the energy-converting device, re-absorption is greatly reduced. Compared to conventional LSC, escape-cone losses can be reduced by more than a factor of two resulting in <10% photons escaping the total internal reflection waveguiding. The concept also allows for stack structures, covering several spectral ranges with ideal pigments and photovoltaics for each range (Fig. 1e) to ultimately convert the entire spectrum with high efficiency.

## Results

**Construction of artificial light-harvesting antennas.** To construct a solar concentrator with randomly oriented donors and aligned single acceptors practically, we first screened systematically standard fluorescence dyes in different polymer materials for their capability to be aligned by stretching the polymers with the pigments included. Using polarized fluorescence spectroscopy, we observed that some pigments undergo a very efficient re-orientation during this process while others did not at all (Supplementary Note 1, Supplementary Table 1, and Supplementary Figure 1). Intriguingly, drastic differences in this behavior were observed even within the same class of fluorescence dyes. For example, Coumarin 6 displayed a very efficient alignment in this process while reorientation of Coumarin 1 was still almost negligible even at 400% polymer foil extension. We suspect that this observation is related to different molecular structures that are either more bulky or elongate. The finding that some fluorescent dyes reorient while others do not put us in the very favorable position to select dyes with spectral ranges matching ideally for efficient Förster energy transfer and superb photovoltaic materials as well as desired donor and acceptor orientations. For a first proof of principle of the concept, we chose Coumarin 1 as light-harvesting and funneling donor and Coumarin 6 as the light-directing acceptor. Both have an excellent spectral overlap and the emission of Coumarin 6 matches with a maximum at ~520 nm perfectly highly efficient InGaP photovoltaic elements. Testing various concentrations and donor to acceptor ratios, we found a ratio of about 10 donor molecules per acceptor and a nearest donor–acceptor inter pigment distance of about ~1.5–2 nm in ~50 μm thick polyvinyl alcohol (PVA) foils stretched by 400% suitable for a first test (Supplementary Note 1 and Methods).

**Goniophotometry.** To assess the orientation and emission characteristics of donor and acceptor molecules, we first detected the three-dimensional (3D) angle-dependent distribution of the absolute intensity emitted by donors and acceptors using a calibrated power meter (photogoniometer, Fig. 2a–f, Supplementary Note 2). Exciting directly the acceptor molecules in a non-oriented sample confirmed an unfavorable preferential emission of the fluorescence light back in directions rather parallel to the excitation (Fig. 2d). This well-known phenomenon is due to photoselection of molecules having their transition dipole moment oriented in the same direction as the polarization of the exciting light (see also Fig. 1c). We observed that light emitted perpendicular to the excitation light, which is the preferred orientation for solar concentrators, was about 20% less intense than light emitted in a parallel direction as the excitation. Next, we tested with the non-oriented sample whether acceptor light was also observable after selective donor excitation and how the emission was distributed at angles nearly perpendicular to the donor excitation. Indeed efficient energy transfer was observed (Fig. 2e). The emission characteristics of the acceptors demonstrated a nearly perfectly isotropic angle distribution providing evidence that any preference in the orientation of excited donors was not transferred to the orientational distribution of excited acceptors. Note that also less photoselection was expected to be seen in the residual donor emission in Fig. 2b as donor excitation was nearly perpendicular to the observation plane (Fig. 2a). Next, we repeated the measurement using a foil expanded by 400%. While the residual donor emission showed only very little reorientation (Fig. 2c), the acceptor emission was clearly very anisotropic (Fig. 2f, Supplementary Figure 2). This provides evidence that the donor molecule orientations largely remained random, still allowing for collection from all incident angles, while the

acceptor molecules were very efficiently aligned. In light concentrators based on silicate glass or poly(methyl methacrylate) (PMMA; refractive index ~1.5), photons that hit the surface to air at angles >48° will not be totally reflected by the wave guide and

are lost (Supplementary Note 2). Therefore, the range that keeps photons in total reflection waveguides corresponds to angles within 90° ± 48° and within 270° ± 48° in Fig. 2e–f. The respective data demonstrate that ~90% of the emitted photons were re-

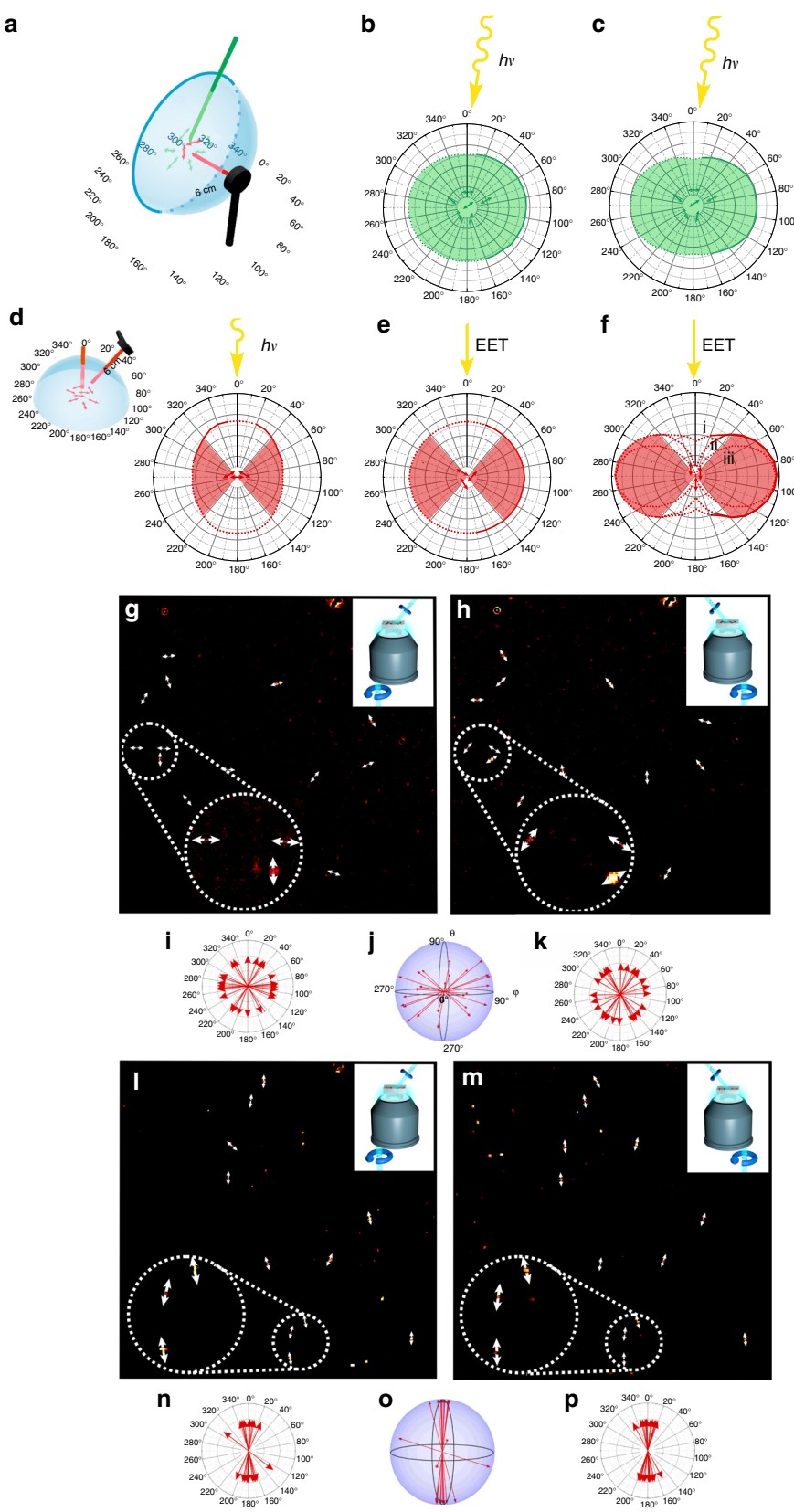

directed into angle ranges suitable for total internal reflection waveguiding (Supplementary Note 2 and Supplementary Table 2) when considering the full 3D angle space (Supplementary Note 2 and Supplementary Figure 3). Higher re-direction efficiencies can be achieved with materials of higher refractive index. For example, for a refractive index of 1.9 the corresponding angle ranges are within $90° \pm 58°$ and $270° \pm 58°$.

**3D single-molecule orientation**. To further confirm that the acceptor molecules were in parallel alignment in all three dimensions, we explored foils at low concentrations that allow for single-molecule studies. Figure 2g, h, l, m shows background-corrected signals of single randomly oriented acceptor molecules (Fig. 2g, h) and of aligned acceptor molecules (Fig. 2l, m, Methods section). The 3D orientation was determined using polarization excitation modulation from two different, tilted incident angles (see insets in Fig. 2g, h, l, m, and Methods for details). From the corresponding two-dimensional transition dipole moment projections (Fig. 2i, k, n and p), the distribution of 3D orientations, Fig. 2j and o, for random and aligned acceptors could be computed, respectively (for details, see Methods and Supplementary Figure 4). These results confirmed a very distinct orientational alignment of the acceptor molecules in stretched foils in all three dimensions (Fig. 2o) while non-expanded foils show a nearly isotropic molecular orientation distribution (Fig. 2j). Note that the angle distribution in more concentrated samples may vary.

**Combined energy transfer and fluorescence quantum yield**. The funneling and re-emission quantum efficiency depends on the combined quantum efficiency for energy transfer and fluorescence quantum yield of the acceptor. The Förster radius of Coumarin 1 and Coumarin 6 is ~5 nm (Supplementary Note 3). Thus high energy transfer quantum efficiencies are expected for the average shortest distances of ~2 nm between donors and acceptors. In addition, the fluorescence quantum efficiency of Coumarin 6 has been reported to be >80% in rigid polymer matrices, but no exact values were available for Coumrain 6 in PVA[10]. Therefore, the most direct way to determine overall funneling and re-emission quantum efficiency in foils with aligned acceptors are absolute angle-dependent power measurements using calibrated power meters as done for the re-direction efficiency determination in Fig. 2f and considering the amount of light absorbed in the set-up (Supplementary Figure 5) and by the foils itself (Supplementary Figure 6). Indeed, the powers measured for data such as presented in Fig. 2f demonstrated

combined quantum efficiencies for energy transfer and fluorescence quantum yield of up to 83% (Supplementary Note 4 and Supplementary Figure 7).

**Excitation light absorption and emission light re-absorption**. Absorption measurements of the donor–acceptor foils demonstrate that 99% of the excitation light was absorbed by a single foil at an incident angle of 20° (Fig. 2a, Supplementary Note 4). In addition, they also show that <0.5% in the spectral maximum of emitted light was re-absorbed by a single foil (Supplementary Note 4, Supplementary Figure 6, ~525 nm). Considering the high fluorescence quantum yield of Coumarin 6, a large amount of these re-absorbed photons will be re-emitted again in favorable directions and thus the overall re-absorption losses are expected to be low.

**Pump–probe experiments**. Finally, we investigated the energy transfer and re-orientation dynamics in the donor to acceptor pigment pool by ultrafast pump–probe experiments. To fully resolve both, the time necessary for funneling and dipole reorientation within the donor pool and the time for the final energy transfer step from the closest donor molecules ideally oriented to the acceptors, we excited the donors with a polarization perpendicular to the acceptor probe wavelength (Inset in Fig. 3b). For both, a distribution of multiple time constants is expected since various donor to donor and donor to acceptor distances as well as relative orientations are present. We observed a short rise with a time constant of about 20 ps and a longer component rising in more than ~200 ps (green curve in Fig. 3b). In addition, there were even longer rise components present after the pulse to pulse time of 8 μs. None of these rise components were observed with a control containing solely acceptor dyes at the same concentration (red curve in Fig. 3b). Note that the foil data are more noisy than typical solution data because the pump–probe beams can only overlap with the ~30–50 μm thin foils that also needed to be moved using a special sample holder (see inset in Fig. 3b and Methods section). Calculating the time scale for a Förster type of energy transfer from the donor and acceptor spectra (Fig. 3a) and a concentration-based estimate for the nearest donor–acceptor center to center distance of ~2.3 nm results in a theoretical value of 31 ps (Supplementary Note 3). Therefore, we attribute the short time constant to the time scales for the final energy transfer steps from donor molecules most favorably oriented and closest to acceptor molecules (Fig. 3c). Correspondingly, we attribute the longer time constants to the time

**Fig. 2** Goniophotometry and three-dimensional single-molecule orientation measurements. **a–f** Emission angle distributions of polymers with donor–acceptor ratios of ~8:1 and distances of ~1.5–2 nm were determined using calibrated power meters. **b**, **e** Isotropic acceptor emission (red, **e**) was observed after nearly perpendicular donor excitation in foils with random acceptor orientations. **c**, **f** When using samples with aligned acceptors instead, most of the light was emitted perpendicular (red, **f**) to the acceptor dipole moments, while nearly isotropic residual donor emission (green, **c**) demonstrates that the light-harvesting donors were still randomly orientated. Approximately 90% of the acceptor photons were re-directed into an angle range suitable for total reflection optical waveguiding with refractive indices of 1.5 or 1.9 (for details see Supplementary Figure 2, Supplementary Table 2, and Supplementary Note 2). The three acceptor distributions in **f** are obtained after (i) donor excitation and EET to aligned acceptors, (ii) direct excitation of aligned acceptors, and (iii) calculating an ideal $\cos^2$-distribution expected from perfectly aligned molecules. **d** When exciting randomly oriented dyes directly, as in conventional luminescent solar concentrators (cf. Fig. 1c), actually more light is emitted in unfavorable directions parallel to the excitation. Intensities at experimentally inaccessible detection angles were linearly extrapolated and values for angles >180° were mirrored from the values at 0°–180° (dotted lines in **b–f**). In **b**, **c**, **e**, and **f**, the scale denotes the angle with the transition dipole moment vector of aligned acceptors in stretched foils (Fig. 2a). In **d**, the scale denotes the angle with the excitation direction (inset in Fig. 2d). **g–p** Three-dimensional orientations of single acceptor molecules were determined by polarization modulated excitation with two different incident light directions (insets in **g**, **h**, **l**, **m**, Methods). **i**, **k,n**, **p** This yielded two different projections of the transition dipole moment orientation for individual molecules in samples with random (**g**, **i** and **h**, **k**) and aligned (**l**, **n** and **m,p**) acceptors, respectively. **j,o** From the two-dimensional projections **i**, **k** and **n,p**, the distribution of three-dimensional dipole orientations, **j** and **o**, for random and aligned acceptors are computed, respectively. These observations confirmed parallel alignment of the acceptor molecules in stretched foils in all three dimensions

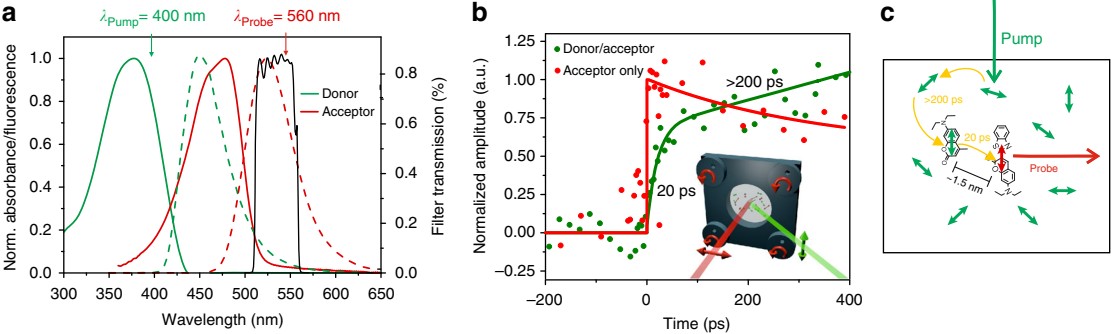

**Fig. 3** Pump–probe measurements **a** Donor (green) and acceptor (red) absorption and emission spectra along with transmission spectrum of the filter used for the power measurements shown in Fig. 2a–f. **b** Transient absorption data (green) observed with a polymer-containing donor ($\lambda_{Exc} = 400$ nm) and acceptor ($\lambda_{Probe} = 560$ nm) of the same ratio (~10:1) and inter-pigment distances (~1.5 nm) as for the data shown in Fig. 2b–f. The polarization of pump and probe beam was perpendicular to cover the entire re-directing and funneling dynamics from randomly oriented donors to perpendicularly oriented acceptors (inset in **b**). The data of the donor–acceptor foil was linearly corrected for a rise component occurring even after the pulse to pulse time of 8 µs (Supplementary Figure 8). Red control: normalized data of a polymer containing only acceptor at the same concentration. **c** We attribute the two observed time scales of ~20 ps and >200 ps to the final nearest donor to acceptor ~1.5–2 nm energy transfer step as well as intra donor pool energy funneling and concomitant dipole re-orientation. Very similar time scales were also observed without linear correction of the long-lasting background. During measurements, the samples were rotated by using a special holder attached to four wheels in a way that guaranteed preserving the orientation of the foils and still keeping the ~30–50 µm thick foils in the pump–probe focus (inset in **b**)

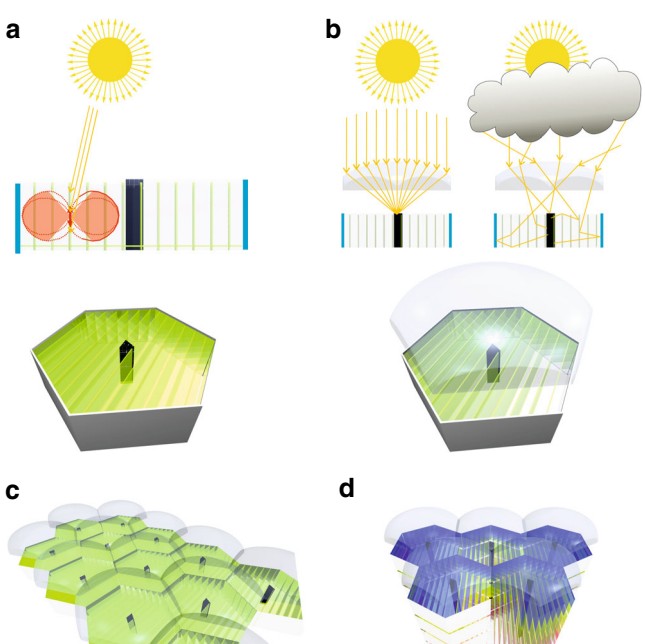

**Fig. 4** Potential photoprotective concentrator architectures. The leaflet-like structure and perpendicular emission of the funneling and re-directing foils is ideal for harvesting architectures using materials of best transparency and waveguiding properties. **a** A possible architecture that allows for large absorption angle ranges, optimized emission angle ranges, and very little re-absorption and reflection losses. The light harvesting to photovoltaics area ratio is about 25:1. **b** Alternative architecture with additional lenses for collecting highly intensive direct sunlight irradiation. This architecture intrinsically switches between efficient light harvesting by photon-redirection of diffuse light under cloudy or shady conditions and direct lens focusing with concomitant pigment protection under intense direct sun irradiation. **c** The hexagonal shape of the proposed architecture allows for larger area arrangements keeping the optical pathways between pigments and photovoltaics small. Electrical access to photovoltaic elements is possible from the bottom. **d** The architecture also allows for stack structures redirecting light to high-efficient photovoltaics for each spectral range

necessary for intra-donor pool energy transfer energy migration toward the acceptor molecules as well as reorientation of the initially unfavorable absorbing donor dipole orientation. We do not know the origin of the very long rise time (>8 µs). In fact, a delayed luminescence could be observed even by eye in the foils. We propose that both might arise from delayed formation of excited singlet states, $S_1^*$. These can be generated by triplet energy migration of several triplet excitons, $T^*$, in the donor pool and a subsequent triplet–triplet annihilation, $T^* + T^* \rightarrow S_0 + S_1^*$, when two such excitons meet[53]. A subsequent energy transfer from such delayed donor singlet states to acceptors potentially contributes additionally to the overall re-direction and energy transfer efficiency and might further explain the high quantum efficiencies that we observed in the absolute power measurements. In summary, the pump–probe data support very efficient energy transfer (Supplementary Note 5). In addition, efficient energy transfer is supported by comparisons of absorption with fluorescence excitation spectra (Supplementary Note 6).

## Discussion

The biomimetic light-harvesting concept for efficient collection, funneling, and re-directing of diffuse light presented here allows for nearly complete re-direction of photons into angle ranges optimized for photovoltaic elements, waveguiding by total internal reflection, and minimizing multiple reabsorption and reflection losses (Figs. 1b, d and 2f). With the flat and leaflet-like structure of the foils, high local donor–acceptor concentrations and distances on length scales of a few nanometers are possible that allow efficient funneling energy transfer and high light-collecting efficiencies (99%, Supplementary Note 4) while re-absorption of the re-directed light is low (<0.5% for a single foil, Supplementary Note 4). The concept also allows for stack structures to ideally cover several spectral ranges and to select ideal pigments and photovoltaics for each range (Fig. 1e). It also allows for extending the spectral collection range to the blue and is extremely cost efficient as PVA, both coumarins, and even silver-coated PMMA are very affordable. We demonstrated the principle with blue photons of ~375 nm and observed an overall light redirection quantum efficiency of ~80% (Supplementary Note 7) at emission wavelengths around 530 nm that are ideally suited for InGaP photovoltaic elements. Note that the quantum

efficiency is the most important parameter as even all high-efficient photovoltaic devices can only convert the energy of each photon corresponding to their own band gap.

The fact that the light is emitted perpendicular with respect to the planes of the foil is advantageous as it allows one to minimize the material necessary for the foils themselves and allows selecting ideal waveguiding material with respect to transparency and refractive index. For example, polymers with a refractive index of >1.9 exist that allow extending the angle range for total reflection up to +/−58°. With the experimentally observed angle distribution, this would decrease escape-cone losses further down to 7–9% and in the ideal case of a $cos^2$ distribution even down to 1–2% (Fig. 2f, Supplementary Note 2, and Supplementary Table 2). Figure 4a–d shows examples for potential practical concentrator architectures that allow for multiple stacking, covering larger areas in a honeycomb-like arrangement and even a self-regulating pigment photo-protection architecture switching between highly efficient collection of intense direct and diffuse indirect light irradiation conditions. The additional use of a lens architecture on the top of the structure can allow for effective photoprotection of the pigments under high intensive direct sun irradiation. Under such conditions, the collimated irradiation will be directly focused by the lens onto the central photovoltaic structure thereby protecting most pigments as they are not excited directly (Fig. 4b). When no direct sun irradiation is present, for example, on a cloudy day or in shady areas, the lens cannot focus the diffuse refracted light directly onto the photovoltaic structure but it will instead be re-directed efficiently by the concentrator sheets. To cover the entire spectral range, the top of the photovoltaic structure could consist of triple junction photovoltaics while the diffuse light still could be collected by stacked structures with photovoltaics optimized for each spectral range (Fig. 4d). Only a small fraction of diffuse light components entering the concentrator vertically would not be absorbed by the vertical foils and could be re-directed at least partially by horizontal foils without aligned acceptors at the bottom (Fig. 4a).

Regarding photo stability, the concept shown in Fig. 4b already demonstrates an architecture that switches automatically between direct lens collection under intensive direct sunlight irradiation and light re-direction using the pigments only under diffuse light irradiation. In addition, advancements in organic light-emitting diode technology have clearly demonstrated that it is possible to achieve lifetimes of organic pigments that easily reach scales of several years if oxygen or other aggressive influences are excluded during the fabrication process. Such measures are also applicable to the approach presented here. Alternatively, more photo stable quantum emitters could be tested[54].

We envision that the great variability of the approach allows for screening many other suitable pigments that have, for example, an ordered transition dipole moment perpendicular to the foil plane, allowing further alternative concentrator architectures and finding acceptors that cover further spectral ranges ideally matching the wavelength of other high-efficient photovoltaic materials. We also envision that the approach presented will not only be used for harvesting diffuse light for high-efficiency photovoltaics but also in multiple other applications, such as all optical logic circuits.

## Methods

**Sample preparation**. First, a homogeneous PVA solution with a PVA concentration of 18 wt% was either obtained by heating PVA (vh, lw) with 10 ml of a mixed water/dimethyl sulfoxide (DMSO) (20%/80%) at 140°C for 2 h in $N_2$ atmosphere[55] or by heating PVA (th, hw) with 10 ml double distilled water under stirring the mixture at 90°C for 2 h in $N_2$ atmosphere[55]. Simultaneously, dye solutions were either prepared by dissolving 0.0035 g of the acceptor Coumarin 6 and 0.0175 g of the donor Coumarin 1 in 10 ml water/DMSO (20%/80%) or by dissolving 0,00175 g Coumarin 6 and 0,0175 g Coumarin 1 in 10 ml Ethanol

(EtOH). After lowering the temperature of the PVA solutions under stirring, the dye solutions were added. Next the solutions were sonicated for degassing. For the preparation of the foils, 2–3 g of the mixtures were casted on a glass plate and allowed to stand after some slewing in a drying cabinet for 2 days. Stretching of dye-doped foil by 400% oriented the acceptors while the donors kept random orientation. The thickness of the resulting foils was on the order of 20–50 μm, depending on the degree of stretching. For the measurements shown in Fig. 2a–f, preparations obtained from the mixed water/DMSO solutions were used while the data observed in Figs. 2g–q and 3 were observed from preparations originally dissolved in water and EtOH.

**Angle-dependent absolute power measurements**. A 375 nm and 485 nm picosecond pulsed diode laser (LDH-P-C 375 and LDH-P-C 485 both from PICOQUANT) were used as light sources. The 375 nm laser was used to excite donor molecules, the 485 nm laser for direct excitation of acceptor molecules. Dependent on the utilized laser, a donor short pass excitation filter (FES0450 from THORLABS) or a 488 nm laser line clean-up filter was installed. A $\frac{\lambda}{4}$-wave plate (achromatic $\frac{\lambda}{4}$-plate, 400–700 nm, NEWPORT CORPORATION) was used to generate circularly polarized light for un-polarized excitation. The foils were directly attached to the center of a glass hemisphere (soda-lime glass from SCHÄFER GLAS, diameter 7.98 cm) using immersion oil to suppress the influence of refraction effects occurring at foil air interfaces (Fig. 2a). The laser excitation was directed at angles of $\Theta = 20°$ or $\Theta = 90°$ onto the foil by using mirrors (angles are defined as in Supplementary Figure 3). The foils with aligned acceptor molecules were attached either with vertically or horizontally aligned transition dipole moments to observe the data shown in Fig. 2 and Supplementary Figure 2. The foils were kept in position using a plano-convex lens (LA1951-B—N-BK7 from THORLABS) in a cage plate (CP02T from THORLABS) at the center backside of the hemisphere. Emission light was detected using a power meter (LabMax-TO from COHERENT, sensitive area of detection: 0.49 $cm^2$). The sensitive area was positioned at different angles, φ, around the equator of the glass hemisphere at a distance of 6.1 cm from the irradiated foil. In front of the detector, a donor band pass (FB450-40 from THORLABS) or acceptor band pass emission filter (ET535/50 m from CHROMA TECHNOLOGY, Fig. 3a) was installed.

**Three-dimensional single-molecule orientation measurements**. For the single-molecule measurements, the sample preparation as described above was used with an acceptor molecule concentration of only $2×10^{-11}$ M. These foils were then bonded with Entellan (MERCK) between cover slips, which were previously cleaned in an ultrasonic bath (2510MT BRANSON) using 2 M lime potash (purchased from SIGMA ALDRICH) and double-distilled water for 10 min. In addition, the cover slips were washed with EtOH and dried using a nitrogen flow directly before usage.

The single-molecule polarization set-up has been described previously[56]. It was extended here for measuring the full 3D orientation of the molecules by irradiating the samples from two different incident light directions (see insets in Fig. 2g, h, l, m). Briefly, a 488 nm continuous-wave (CW) laser (sapphire 488–50, COHERENT) was used for excitation. At first, the beam passed a telescope system (achromatic doublets, $f = 30$ mm and $f = 500$ mm, THORLABS). Next, the beam passed a dichroic mirror (beamsplitter z 568 sprdc, AHF). In addition, a rotating $\frac{\lambda}{2}$-wave plate (achromatic $\frac{\lambda}{2}$-plate, 400–800 nm, THORLABS) led to a continuous rotation in the polarization vector. The rotation was achieved through a chopper wheel (Optical Chopper System, THORLABS), which was synchronized to the electron-multiplying charge-coupled device (EMCCD) camera (iXonEM+897 back illuminated, ANDOR TECHNOLOGY). Subsequently, the laser beam passed two wedge-prisms (4° Beam Deviation, 375–700 nm, THORLABS), which were used to shift the beam laterally for the two positions at the back aperture of the microscope objective (insets in Fig. 2g, h, l, m). After that, the beam passed a third lens (Achromatic Doublet, $f = 500$ mm, THORLABS) before reaching the microscope. Through a further dichroic mirror (beamsplitter z 488 RDC, AHF) the beam was focused onto the back aperture of the objective at the two different positions (NA = 1.35 oil immersion objective lens, UPlanSApo, 60×, OLYMPUS). Emission light passed through previously mentioned dichroic mirror, two emission filters (ET band pass 525/50, AHF and BrightLine HC 525/30, AHF) and a further telescope system (achromatic doublet, L4: $f = 60$ mm, L5: $f = 250$ mm, NEWPORT CORPORATION) to focus the image on EMCCD camera.

The laser output power was 8.4 mW and the electron-multiplying gain was 300. In two measurement directions with maximum distance, 700 frames were recorded with a frame rate of 30 Hz. One period of excitation polarization rotation corresponded to 15 frames. During the first 200 frames, a polarization filter was set in optical path to determine the absolute orientation of the polarization for each frame.

To remove the large background emission from the PVA polymer in the single-molecule detection experiments, the following procedure was done. First, 285 frames of the videos were averaged to yield a single 15 frame movie corresponding to a single averaged period of polarization orientation. To minimize the background, first a Gaussian blur of radius 8 (corresponding to Sigma 8 in units of pixels) was applied to each of the 15 frames using the software package ImageJ. These Gaussian blurred images were then subtracted from the corresponding single period average. In addition, the videos were pixel-binned 2 × 2. Finally, average

intensities were generated and molecules were visualized using the "Red Hot" lookup table from ImageJ and displaying only positive values to obtain the data shown in Fig. 2g, h, l, m.

Next, various molecules that were clearly visible in both directions of incident light were selected using rectangular regions of interest (ROIs). For both videos, the modulation phase of every ROI was calculated using fast Fourier transform. The offset of the phases were corrected using the calibration with the polarization filter as well as divided by two to transform the phase into an orientation angle. For each ROI, the two different directions of incident light yield two angles $\alpha_1$ and $\alpha_2$ that represent the orientation of the projection of the transition dipole moment vector onto the plane perpendicular to the respective incident light propagation directions (Supplementary Figure 4). These angles were used to derive the spherical, 3D transition dipole moment orientation (coordinates $\varphi$ and $\theta$ for the respective molecule using the following formalism). First, the vector orientation of the dipole moment (green in Supplementary Figure 4) in Cartesian coordinates, $x$, $y$, $z$, was obtained using the following equation, with $\beta$ being the absolute angle between the optical axis ($z$) and the two propagation directions of the incident light, respectively.

$$\vec{r} = \begin{pmatrix} x \\ y \\ z \end{pmatrix} = \begin{pmatrix} \cos\alpha_1 \\ -\frac{1}{2}\sec\beta\sec\alpha_2\sin(\alpha_1+\alpha_2) \\ \frac{1}{2}\csc\beta\sec\alpha_2\sin(\alpha_1-\alpha_2) \end{pmatrix} \quad (1)$$

This equation was derived by calculating the intersection line of the two planes spanned by the respective polarization vector of the incident light and the vector describing its respective direction of propagation. After normalization of the vector $\vec{r}$ to unity length, the spherical coordinates were calculated as follows (using the now normalized $x$, $y$, and $z$ coordinates, denoted by index $n$)

$$\varphi = \operatorname{atan2}(-y_n, x_n) \quad (2)$$

$$\theta = \arcsin(z_n) \quad (3)$$

The atan2 function is the arctangent function with two arguments, assigning the correct quadrant to $\varphi$ depending on the signs of $x_n$ and $y_n$. The negative of $y_n$ was used to consider the direction of rotation of the incident light.

By this method, the full 3D orientation of a molecule can be reconstructed unless it is oriented in the $yz$-plane (the plane spanned by the two directions of incidence of the excitation light), i.e., $\varphi = 90°$ or $\varphi = 270°$. In this exceptional case, $\alpha_1$ and $\alpha_2$ are both equal to 90°, which results in an indeterminate $z$-component of the orientation vector $\vec{r}$, thereby rendering the calculation of the molecule's out-of-plane angle $\theta$ impossible.

Please note that the distribution of randomly oriented molecules in Fig. 2j is not perfectly isotropic since molecules with transition dipole moments perpendicular to the surface of the microscope objective are significantly harder to detect as they emit most of the light in directions parallel to the surface.

**Pump–probe measurements**. To minimize singlet–singlet annihilation effects, a high-repetitive laser system (Coherent OPA/Rega operated at 120 kHz) with low per pulse energies was used for the pump–probe experiments. The pump–probe set-up has been described previously[57, 58] and was modified for measuring the energy transfer migration kinetics in the foils. In detail, a RegA 9000 was pumped by a Vitesse Duo Laser (components from COHERENT INC.). The laser beam was split within an optical parametric amplifier OPA 9450 (also from COHERENT INC.). In the OPA 9450, a part of the RegA 800 nm was used for white light generation for probing while another part was frequency doubled for the generation of a 400 nm pump beam for exciting the donor molecules. The pump pulse duration was ~160 fs and the pump pulse energy was about ~30 nJ per pulse, corresponding to about $10^{16}$ photons/cm$^2$ or less per pulse. Such intensities are known to result in significantly less than one excitation per ~10 pigments[59]. The probe wavelength of the white light output was selected by a linear gradient filter ($\lambda = 560$ nm). The wavelength was adjusted by displacement of the linear gradient filter and by using a miniature spectrometer USB2000+ (OCEAN OPTICS). Emission light was detected using an ultrafast photodiode and a lock-in amplifier (EG&G 5205). The signal of the lock-in amplifier was coupled with a mechanical chopper positioned in the optical path of the pump beam. The pump beam passed a short pass filter (AHF HC770/SP) and a motorized linear stage, which had a travel range of 30 cm. From here, the beam was focused by an achromatic lens ($f = 50$ mm) onto dye-doped polymer foils, which were fixed with a special sample holder. To minimize photo bleaching of the samples, the special holder was attached to four wheels in a way that guaranteed preserving the orientation of the foils during rotation and still keeping the ~30–50 µm-thick foils in the pump–probe focus (inset in Fig. 3b). The probe beam also passed a short pass filter (AHF HC770/SP) and an additional linear gradient filter. After that, the probe beam was also focused onto dye-doped layer in the same spot as the pump beam. The pump beam was polarized vertically while the probe beam was polarized horizontally (inset in Fig. 3b). Please note that similar kinetics are expected with oriented as well as non-oriented acceptors since also in the case of randomly oriented acceptors first intra donor pool energy transfer to a donor molecule with favorable distance and

orientation must occur, irrespective of the orientation within the laboratory frame. After the sample, the probe beam was directed into a Czerny-Turner Spectrometer (from CHROMEX) for further probe wavelength selection and detected by a fast photodiode (design by Professor D. Schwarzer, Göttingen). The very long-lived rise component in the donor–acceptor system was considered by fitting a linear background correction in addition to fitting a bi-exponential rise (Supplementary Figure 8).

**Data availability**. Most of the relevant data are provided in the Supplementary Information. All other relevant data are available from the authors upon request.

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

## Acknowledgements

This work was financially supported by the Deutsche Forschungsgemeinschaft (DFG) (INST 188/334-1 FUGG).

## Author contributions

A.P. and P.J.W. designed experiments, analyzed data, and wrote the paper. A.P. and P.J. W. developed the protocol to generate the samples. A.P., M.W., and M.H. generated samples and performed goniometric measurements. Single molecule experiments were done by M.H. and D.P. and pump–probe measurements by M.W., D.A.G., and N.H. A.A. analyzed data. All authors edited the manuscript.

## Additional information

**Competing interests:** The University of Braunschweig and two authors (A.P. and P.J. W.) filed a patent for parts of this work. The remaining authors declare no competing financial interests.

