## [Peer Review File · Nature Communications]

Reviewers' comments:

Reviewer #1 (Remarks to the Author):

The manuscript by Pieper et al. is an attractive and very interesting story aiming for construction of directional light collectors based on organic dyes embedded in polymer foil. It describes a novel approach that can provide a basis for future expansion of this interesting idea. Though it is too early to conclude whether the devices constructed by the authors can compete with current commercial products, the FunDiLight concept obviously has a potential to be an attractive model for constructing light-harvesting systems for light-driven nanodevices.

I especially value the effort of authors to present the data in a very clear and concise way, the supporting material show the raw data and supplementary notes contain useful information for any followers that would want to repeat the experiments. The presented data demonstrate convincingly that the random donor - oriented acceptor system in a stretched foil indeed direct the absorbed diffuse light into a preferential direction.

The only drawback of the presented concept is the absorption range of the selected donor, Coumarin 1, which has a maximum at 375 nm, thus in wavelength range a bit too short to be used in commercial systems for collecting sunlight. I understand that the goal was to have the emission range falling into the high sensitivity of InGaP detectors, but essentially zero absorption of the acceptor above 430 nm makes it inappropriate for the use as a sunlight collector. Yet, the manuscript provides a proof of principle that the concept developed by the authors works and further improvement will be needed to make it suitable for collecting sunlight. I am positive that this work will have a significant impact in the field of photovoltaics and artificial photosynthesis.

A question that should be explained in the revised version: In Fig. 2, the authors show that direct excitation of the randomly oriented acceptors (Fig. 2d) there is preferential emission in the direction parallel to the excitation. Why then the excitation of randomly oriented donors (Fig. 2b) is isotropic? Either it is wrongly explained in the text and figure caption, or there is some conceptual mistake.

Reviewer #2 (Remarks to the Author):

In "Bio-mimetic light-harvesting funnels for re-directioning of diffuse light", the authors present a new approach to collect and transduce absorbed photoenergy for applications such as solar energy devices. This approach, termed FunDiLight (funneling diffuse light redirectioning), combines isotropic donors with oriented acceptors in a polymer foil. Upon expansion of a polymer foil, some dyes align with the expansion (and from these dyes the acceptors are chosen) while others do not (and from these dyes the donors are chosen). In this material, the donors absorb incoherent sunlight and transfer the absorbed energy to the acceptors. The energy then migrates between the acceptors to reach the edge of the foils, where it can excite an attached PV. The authors perform careful measurements to show an ~80% quantum efficiency associated with energy transfer and fluorescence remission for their material. Additionally, they perform pump-probe measurements to demonstrate donor to acceptor energy transfer.

The material introduced here leverages a clever idea to use polarization to separate and extract photoenergy from the absorbed material. The new material is simple to construct and interesting, and so the work should certainly be published. However, prior to publication the authors may want to consider the points below.

1. The authors' state that their material allows ~90% of the acceptor photons to be emitted at

angles acceptable for TIR of materials with $n=1.5$. To illustrate the advantage of their material, they should clearly state the percentage of isotropic photons that are emitted at acceptable angles. Would it be 56% (that is, for 90 ± 50 degree range)?

2. There are a few aspects of the pump probe measurements that would benefit from clarification,

a. The authors perform cross-polarized pump-probe measurements to look at orientational changes. Pump-probe anisotropy measurements would better separate the orientational dynamics.

b. Did the authors perform power dependence measurements to rule out annihilation effects?

c. What are the pulse characteristics (energies, pulse duration, etc.)?

We are very grateful for the reviewers' positive and constructive remarks that helped us significantly improving the manuscript. In the following we have addressed all comments in a point by point manner. In addition, we have also addressed all format requirements. In particular, we have significantly shortened the abstract. All additions in the manuscript are marked by bold letters while deletions are marked by deletion lines.

Reviewer #1 (Remarks to the Author):

The manuscript by Pieper et al. is an attractive and very interesting story aiming for construction of directional light collectors based on organic dyes embedded in polymer foil. It describes a novel approach that can provide a basis for future expansion of this interesting idea. Though it is too early to conclude whether the devices constructed by the authors can compete with current commercial products, the FunDiLight concept obviously has a potential to be an attractive model for constructing light-harvesting systems for light-driven nanodevices.

I especially values the effort of authors to present the data in a very clear and concise way, the supporting material show the raw data and supplementary notes contain useful information for any followers that would want to repeat the experiments. The presented data demonstrate convincingly that the random donor - oriented acceptor system in a stretched foil indeed direct the absorbed diffuse light into a preferential direction.

The only drawback of the presented concept is the absorption range of the selected donor, Coumarin 1, which has a maximum at 375 nm, thus in wavelength range a bit too short to be used in commercial systems for collecting sunlight. I understand that the goal was to have the emission range falling into the high sensitivity of InGaP detectors, but essentially zero absorption of the acceptor above 430 nm makes it inappropriate for the use as a sunlight collector. Yet, the manuscript provides a proof of principle that the concept developed by the authors works and further improvement will be needed to make it suitable for collecting sunlight. I am positive that this work will have a significant impact in the field of photovoltaics and artificial photosynthesis.

A question that should be explained in the revised version: In Fig. 2, the authors show that direct excitation of the randomly oriented acceptors (Fig. 2d) there is preferential emission in the direction parallel to the excitation. Why then the excitation of randomly oriented donors (Fig. 2b) is isotropic? Either it is wrongly explained in the text and figure caption, or there is some conceptual mistake.

We thank the reviewer to make us aware that this point might have been unclear. First of all, from the comments we realized that the orientation of the half sphere sketched in Figure 2a can actually been seen in two different ways (like a multistable perception such as the necker cube) – one in which the flat side points to the upper left of the page and one to the back of the page, with the latter being the correct one. A quick survey asking several people showed that, unfortunately, a major fraction of people see the flat side pointing to the upper left, which is the wrong orientation. This might have contributed to some confusion. We now have tried to eliminate that ambiguity in the figure and made the orientation of the sphere as well as excitation and detection angles more clear:

Old Figure 2 a:

New Figure 2 a:

One reason for the different anisotropies in Figure 2b and d are different excitation geometries. In the measurement of the donor-acceptor systems (Figure 2 b,c,e and f) their ability to redirect light into perpendicular directions was assessed by donor excitation nearly perpendicular to the plane of detected emission angles (green excitation line in Figure 2a, see also supplemental Figure 2). In contrast, the experiment confirming photoselection (Figure 2d) can only be done with the (acceptor) excitation vector being within the plane of detected emission angles, i.e. detection angles including also nearly parallel detection angles. Otherwise, no photoselection can be detected - photoselection is always isotropic in a plane perpendicular to the excitation. Therefore, it is clear that photoselection of the residual emission of the donors shown in 2 b is much less visible, as in these experiments excitation was nearly perpendicular to the plane of detected emission angles. In addition, energy transfer within the higher concentrated donor pigment pool can also decrease initial photoselection.

To further clarify these differences we added in line 157

“Next, we tested with the non-oriented sample whether acceptor light was also observable after selective donor excitation and how the emission was distributed at angles nearly perpendicular to the donor excitation.”,

and in line 164

“Note that also less photoselection was expected to be seen in the residual donor emission in Figure 2 b as donor excitation was nearly perpendicular to the observation plane (Figure 2 a).”

and added an inset into figure 2d that further illustrates how the photoselection of directly excited pigments is detected relative to the excitation.

Reviewer #2 (Remarks to the Author):

In “Bio-mimetic light-harvesting funnels for re-directioning of diffuse light”, the authors present a new approach to collect and transduce absorbed photoenergy for applications such as solar energy

devices. This approach, termed FunDiLight (funneling diffuse light redirectioning), combines isotropic donors with oriented acceptors in a polymer foil. Upon expansion of a polymer foil, some dyes align with the expansion (and from these dyes the acceptors are chosen) while others do not (and from these dyes the donors are chosen). In this material, the donors absorb incoherent sunlight and transfer the absorbed energy to the acceptors. The energy then migrates between the acceptors to reach the edge of the foils, where it can excite an attached PV. The authors perform careful measurements to show an ~80% quantum efficiency associated with energy transfer and fluorescence re-emission for their material. Additionally, they perform pump-probe measurements to demonstrate donor to acceptor energy transfer.

The material introduced here leverages a clever idea to use polarization to separate and extract photoenergy from the absorbed material. The new material is simple to construct and interesting, and so the work should certainly be published. However, prior to publication the authors may want to consider the points below.

1. The authors' state that their material allows ~90% of the acceptor photons to be emitted at angles acceptable for TIR of materials with $n=1.5$. To illustrate the advantage of their material, they should clearly state the percentage of isotropic photons that are emitted at acceptable angles. Would it be 56% (that is, for 90+/-50 degree range)?

The fraction of photons that are emitted at acceptable angles from normal isotropic or photoselected sources has been discussed in the literature in quite some detail (see for example McDowall et al, "Comprehensive analysis of escape-cone losses from luminescent waveguides", Ref. 21). Therefore, we are grateful for the reviewers' comment as it inspired us to include a more detailed discussion of this literature in comparison to our material. Typically, the literature gives the percentage of photons emitted at angles that are lost and calls this quantity escape cone losses. The escape-cone loss with isotropic emission is at least about twice as high as in our current material when assuming a refractive index of $n=1.5$. When additionally considering that in conventional light-concentrators photoselection can cause even more escape-cone losses (due to preferential re-emission parallel to the excitation, compare Figure 1 c and d as well as 2 d and f) then the factor in the improvement is even higher than that. However, the overall percentage of photons re-emitted in acceptable angles is above 56 %, even when considering photoselection of randomly oriented fluorophores (see again McDowall et al, "Comprehensive analysis of escape-cone losses from luminescent waveguides"). To make this clear we added in the introduction in line 86:

"Besides reabsorption these photon escape-cone losses often dominate all other loss mechanisms in conventional luminescent solar concentrators (LCSs). For randomly oriented molecules and considering photoselection, 30 % of the light is lost due to escape-cone losses and the percentage for perfect isotropic emission is only a few percent smaller (McDowall et al)."

And In line 120

"Compared to conventional luminescent solar concentrators escape-cone losses can be reduced by more than a factor of two resulting in less than 10 % photons escaping the total internal reflection waveguiding."

2. There are a few aspects of the pump probe measurements that would benefit from clarification,
- The authors perform cross-polarized pump-probe measurements to look at orientational changes. Pump-probe anisotropy measurements would better separate the orientational dynamics.
 - Did the authors perform power dependence measurements to rule out annihilation effects?
 - What are the pulse characteristics (energies, pulse duration, etc.)?

We are also grateful for these useful suggestions and added the requested pulse characteristics in the materials and methods section and also addressed the comment on annihilation effects:

“To minimize singlet-singlet annihilation effects a high-repetitive laser system (Coherent OPA/Rega operated at 120 kHz) with low per pulse energies was used for the pump-probe experiments.”

“The pump pulse duration was ~160 fs and the pump pulse energy was about ~30 nJ per pulse, corresponding to about 10^{16} photons/cm² or less per pulse. Such intensities are known to result in significantly less than one excitation per ~10 pigments. . (Barzda et al., Singlet–Singlet Annihilation Kinetics in Aggregates and Trimers of LHCII)”

Regarding the pump-probe experiments we first performed cross-polarized pump-probe experiments to explore the kinetics of one of the most important functions of the material - re-orienting photons into a perpendicular direction. We fully agree that alternative variations of the orientations in the pump and probe beams as well as chromophore orientations are interesting in order to further explore the orientational dynamics. However, when considering that the pump as well as the probe beams can be oriented each in three different orientations with respect to the acceptors (parallel, vertical and in magic angle orientation), with each experiment providing different information, and that each experiment can be done with donor or acceptor excitation and/or probing and that additionally all experiments should be repeated with control samples either lacking the donor or acceptor as well as using samples with randomly oriented or aligned acceptors we soon realized that this is a large study on its own with literally hundreds of different possible experiments. Therefore, we concluded that such a large study is beyond the scope of the main proof of principle presented here and should be subject to future studies.

REVIEWERS' COMMENTS:

Reviewer #1 (Remarks to the Author):

In the revised version, the authors clarified the point that has made some confusion in understanding the results. The revised manuscript represents a high quality work with significant potential in the field of artificial light harvesting.

Reviewer #2 (Remarks to the Author):

All my concerns have been satisfactorily addressed, and I recommend for publication.